# Recent Advances of Enzyme-Free Electrochemical Sensors for Flexible Electronics in the Detection of Organophosphorus Compounds: A Review

**DOI:** 10.3390/s23031226

**Published:** 2023-01-20

**Authors:** Gayani Pathiraja, Chartanay D. J. Bonner, Sherine O. Obare

**Affiliations:** 1Department of Nanoscience, Joint School of Nanoscience and Nanoengineering, University of North Carolina at Greensboro, Greensboro, NC 27401, USA; 2Department of Nanoengineering, Joint School of Nanoscience and Nanoengineering, North Carolina A&T State University, Greensboro, NC 27401, USA

**Keywords:** organophosphorus compounds, enzyme-free, electrochemical sensors, flexible electronics, nanomaterials, detection limits, stability

## Abstract

Emerging materials integrated into high performance flexible electronics to detect environmental contaminants have received extensive attention worldwide. The accurate detection of widespread organophosphorus (OP) compounds in the environment is crucial due to their high toxicity even at low concentrations, which leads to acute health concerns. Therefore, developing rapid, highly sensitive, reliable, and facile analytical sensing techniques is necessary to monitor environmental, ecological, and food safety risks. Although enzyme-based sensors have better sensitivity, their practical usage is hindered due to their low specificity and stability. Therefore, among various detection methods of OP compounds, this review article focuses on the progress made in the development of enzyme-free electrochemical sensors as an effective nostrum. Further, the novel materials used in these sensors and their properties, synthesis methodologies, sensing strategies, analytical methods, detection limits, and stability are discussed. Finally, this article summarizes potential avenues for future prospective electrochemical sensors and the current challenges of enhancing the performance, stability, and shelf life.

## 1. Introduction

The increased use and mishandling of pesticides to sustain high food production worldwide has directly contributed to environmental pollution and adverse health effects [1,2]. The U.S. Environmental Protection Agency (EPA) reported the total usage of 1,989,454 tons of pesticides in the United States (U.S.) during 2010–2014, which increased to 2,038,895 tons of total usage in the years 2015–2019 [3]. Among the high annual usage of pesticides in the US, the U.S. Geological Survey (USGS) has reported the most recent data for 2013–2017 [4]. Based on these USGS data, the Earthjustice Council, a nonprofit organization, mapped the widespread use of 17 organophosphorus (OP) pesticides within the U.S. in 2017 to determine the registered crops and high residue food exposure routes and their health effects [5]. These data have shown the country’s higher spread areas of OP pesticides. The National Water-Quality Assessment (NAWQA) Project also has stated the annual organophosphorus pesticide usage separately, and Figure 1 displays their estimation of chlorpyrifos, which is one of the most used pesticides in the U.S., for 2018 [6]. They have been categorized into two methods, EPest-low and EPest-high, according to the different treatment situations. EPest-low considered zero usage of pesticide-by-crop combination in the Crop Reporting Districts (CRD), while EPest-high estimated the unreported usage of pesticide-by-crop combination in the CRD. There is a wide spread of chlorpyrifos in the U.S., although it has shown a decreasing trend of use over the years for crop production. Currently, only preliminary data are available for 2019–2022 [7].

The OP pesticides are non-point source pollutants used as fungicides, herbicides, insecticides, and nematicides to limit crop damage. Unfortunately, less than 5% of these pesticides can reach the intended target to yield crop production [8]. The remaining significant percentage enters the environment and pollutes the water, air, and soil, posing risks to non-target organisms [9,10]. Many OP pesticides can influence biological, chemical, and physical processes once they enter the environment. They can degrade and form intermediate products; be absorbed by other organisms via inhalation, ingestion, and skin penetration; or they leach into ground or surface water in the environment. Therefore, these OP compounds have been classified as persistent pesticides as they can remain in the environment for a long time [11]. There is a greater potential for them to accumulate in different organisms in various ways that cause long term health problems.

The chemical structure of OP pesticides consists of a central phosphorus atom with either a double-bonded oxygen (P=O) or a double-bonded sulfur atom (P=S) and two single-bonded hydrocarbon groups and a leaving group. They are structurally thions and oxons, synthetic amides, esters, and thiol derivatives of phosphoric, phosphonic, and phosphonothioic acids. The most recently used OP insecticides in the U.S. and their chemical structures are summarized in Table 1. Exposure to these OP pesticides presents many health hazards, and they cause significant damage to the nervous system by disrupting the cholinesterase enzyme that regulates acetylcholine. Other symptoms such as headache, dizziness, nausea, vomiting, reduced heartbeat, and diarrhea have been shown, while excess exposure can lead to death. Recent studies have reported that prenatal exposure can elevate neurodevelopmental disorders such as attention-deficit hyperactivity disorder (ADHD) and autism [11].

Due to the widespread use of OP pesticides and their adverse health effects, it is crucial to monitor accumulation in the environment, ecology, and food by detecting their accurate concentration in real time. The most common detection methods are chromatography-mass spectroscopy techniques [12,13,14], immunoassays [15,16], and biosensors [17,18,19,20]. Nevertheless, these approaches have practical limitations such as low sensitivity, lack of portability, limited selectivity, difficulties in real-time monitoring, and operational complexity. Enzyme-based biosensors are mostly studied and deeply understood as catalytic or inhibition biosensors because they exhibit more sensitive, rapid, and feasible features than chromatography-based instruments [21]. In addition, various biomolecules, such as enzymes and nucleic acids, provide selectivity of OP compounds and the opportunity for multi-detection [22,23]. However, the main bottlenecks for the feasibility of enzyme-based biosensors are the higher cost of the enzyme, their limited storge stability due to their denature, and short lifetime [24]. In real samples, enzymes can bind with heavy metals, leading to a lack of selectivity and false positive signals [25]. Many different approaches have been developed to improve the stability of enzymes using genetically modified enzymes [26,27], extremophiles [28,29], or surface reconstruction with nanomaterials [30,31,32]. These enzyme-based sensors still need to improve the intrinsic instability to the environmental conditions and the selectivity. Recently, more attention has been given to developing enzyme-free sensors, which have higher sensitivity and a longer shelf life to overcome the limitations of enzyme-based sensors [33,34,35]. Electrochemical detection has displayed promise as an enzyme-free detection method for environmental contaminants. Due to easy portability, sensitivity, and accuracy, studies have explored options for detecting OP compounds. A key strategy is the utilization of nanomaterials. Various functional nanomaterials of metallic and non-metallic nanomaterials, including metal oxide nanostructures [36,37], metals [38], carbon nanotubes [37,38,39], metal organic frameworks [40], molecularly imprinted polymers [41], and molecules [42,43], have been studied as a selective and highly sensitive sensor for the enzyme-free detection of OPs, by utilizing their high electrocatalytic activity and surface area. These nanomaterials offer numerous opportunities to address miniaturization, portability, and rapid detection. However, it is still necessary to combine them with enzymes or metal nanoparticle-labeled antibodies as recognition elements to detect non electroactive OP compounds when fabricating electrochemical sensors [44]. 

The key requirements for developing sensors in practical applications are higher sensitivity, selectivity, stability, reproducibility, and portability. In addition to the targeting of high sensitivity and selectivity of sensors, there is a need to find novel materials that are robust under changes in environmental conditions, i.e., temperature, humidity, chemicals, and pH. More importantly, because there is a broad range in toxicity levels of OP pesticides, it is necessary to develop new methods to discriminate between various OP pesticides. The portability also should be addressed by integrating them into portable devices. The progress in the development of miniaturized sensors using nanomaterial-modified interfaces, such as screen-printed electrodes and photolithographically fabricated electrochemical sensors, has opened the development of portable sensors for the detection of organophosphorus pesticides [22,45,46]. Recently, portable label-free fluorescence probes based on nanomaterials have been successfully fabricated for the rapid and onsite detection of pesticides [47]. The real-time sensors should also be addressed to reduce or minimize false positives, which enhances their applicability. Dual signal transduction provides a more robust detection method and can limit false positives by relying on two separate testing methods when analyzing samples [48]. Our group has successfully demonstrated such developments of independent electrochemical molecular sensors using phenanthroline derivatives and azastilbene [42,43]. 

Exceptional devotion must be given to translating existing primary electrochemical sensing electrodes into wearable, flexible devices specially designed to detect OP compounds in the environment. Although there are several wearable electrochemical devices on the market, there needs to be a reported fully integrated system to track the levels of OP. Therefore, a significant research endeavor has focused on the fabrication of flexible electronic devices due to their advantages such as being lightweight, disposable, and able to couple with other modern devices [49,50]. These devices must retain both mechanical compliance and electrochemical activity, which will provide the convenience to use them without any issues of contamination and recalibration. However, most of the metal deposition methods, such as chemical vapor deposition, physical vapor deposition, sputtering methods and photolithography, are generally expensive. Therefore, it is critical to develop cost effective methods by selecting suitable materials and fabrication methods while retaining their intrinsic properties to reduce the cost of the sensor [49,51]. Miniaturization has always improved the signal-to-noise ratio to provide higher sensitivity, better efficiencies due to reduced non-specific binding, and lower assay costs due to the smaller sample volumes. Therefore, tremendous efforts are needed to review scaling up electrochemical sensors that have the potential for the integration of flexible devices [52]. These flexible and portable sensors are beneficial to improve the health concerns and safety of the environment by detecting OP pesticide residues in real samples such as food and water.

This review intends to highlight the remarkable advancement in the fabrication of various novel materials without the support of enzymes fabricated for electrochemical sensors. While giving a comprehensive overview of non-enzyme-based electrochemical sensors, the present work intends to give a flavor of the miniaturization of these sensors into portable and disposable devices. The electrochemical sensing mechanisms with examples, analytical methods, nanomaterial synthesis methods, their properties, and detection limits are described in detail. The most exciting achievements and the current state-of-the-art of flexible electronic sensors using nanomaterials and molecules for OP detection with their limits of detection are discussed. The importance of the integration of electrochemical sensors into flexible electronics with existing examples, their limitations, and challenges are thoroughly reviewed. Further, future perspectives and challenges of real-time applications of flexible electrochemical sensors are addressed.

## 2. Electrochemical Sensors

The development of electrochemical sensors for the rapid detection of pesticides is growing, with the strong driving forces of their intrinsic benefits of cost, sensitivity, speed, simplicity, and sensitivity. More importantly, these electrochemical sensors are easier to miniaturize and integrate into flexible electronics for onsite monitoring of pesticide traces, compared to other chromatography and spectroscopic methods. Non-enzyme-based electrochemical sensors have become a more popular technique in recent years due to their low cost, high sensitivity, and remarkable stability under different environmental conditions, overcoming the limitations of nanozyme or enzyme-based sensors [25,33,38]. Signal detection can be amplified by using different microstructures, nanomaterials, and molecules as electron carriers to electrochemical interfaces in the electrochemical sensor. The electrode modification strategies are a key factor to enhance the electrochemical properties of the sensor, with them having high electrocatalytic activity, porosity, large surface areas with interconnected pores, and intrinsic conductivity [53].

### 2.1. Working Principle of an Electrochemical Sensor

The electrochemical sensor is composed of a receptor, a transducer, and a detector to transfer the chemical changes involving charge transfer reactions into an electrical or other processable signal, as shown in Figure 2. The OP pesticide and the analyte can selectively interact with the receptor element in the electrode. The receptor can be fabricated using different nanomaterials, such as metals, metal oxides, graphene, or carbon nanotubes, or micro- or nanostructures such as metal organic frameworks, molecularly imprinted polymers, and molecules. The most commonly used electrodes are carbon paste, glassy carbon, graphite, boron-doped diamond, gold, silver, or platinum electrodes. Among them carbon-based electrodes are a more promising candidate for the preparation of low-cost sensors [54,55]. These enzyme-free sensors perform by generating an electrical signal that correlates with the concentration of the analyte compound. There are different detection principles available in electrochemical sensing, including voltammetry, amperometry, potentiometry, and electrochemical impedance spectroscopy (EIS) [56]. Voltammetry is an active technique to measure the current density of an electrochemical process at a constant or varying applied potential, while amperometry quantifies the current intensity generated at a constant potential with reduced or oxidized species of an electroactive species in the solution [57,58]. These methods provide a method to distinguish electrochemical systems by calculating the number of electroactive species of the process as the measured current is directly proportional to the concentration of the analyte [58]. In contrast, potentiometry estimates the applied potential measurements between the working electrode and a reference electrode that can quantify the concentration of the analyte [57]. EIS measures the electrode impedance determined by applying a potential modulation and detecting the current response at different frequencies [59].

### 2.2. Nanomaterial-Based Sensors

Nanomaterial-based sensors play a significant role in improving the sensitivity of an electrochemical sensor to detect OP pesticides. The properties of nanoscale materials are totally different from bulk materials due to the quantum confinement effect. The interesting physicochemical properties of nanomaterials such as the size, morphology, composition, high surface area to volume ratio, and porosity amplify the electrochemical signals of the sensor [53,60,61]. The synergic effect of the improved conductivity and catalytic activity of nanomaterials also improves the efficacy for better performances of electrochemical sensing, which leads to various environmental applications [54]. Therefore, nanomaterials with great conducting properties, such as metal, metal hydroxide/oxide nanostructures, graphene, carbon nanotubes (CNTs), carbon nanofibers, and functionalized polymeric materials, have been widely used for enzyme-free electrochemical sensors. The surface of the electrode can be modified by simply coating these nanomaterials that can be attached to the electrode surface by non-covalent forces or covalent modification [62]. Moreover, the surface coating of materials can be performed by different strategies, such as layer-by-layer assembly [63], electrophoretic deposition [64,65], spin coating [66], dip coating [67], or electrospinning [68]. When the modified surface forms a thin film, it provides a higher conductivity and stability that leads to increased sensitivity for the lifetime of the electrochemical sensor. Improper coating protocols restrict the charge transfer reaction between the electrode and the electrolyte and cause poor sensitivity of the sensor. 

Table 2 summarizes the nanomaterial-based electrochemical sensors reported for the detection of OP compounds. Among the different metallic nanostructures, silver (Ag), gold (Au), and palladium (Pd) nanoparticles (NPs) are the only examples reported as either hybrid structures or nanocomposites that bind with metal oxides or CNTs or graphene-like materials [44,69,70,71,72]. These expensive noble metals have extraordinary catalytic activity and conductivity, which can enhance the sensitivity and stability of the sensor. Metal oxide nanostructures, including zirconia (ZrO_2_), nickel (II) oxide (NiO), copper (II) oxide (CuO), and titanium dioxide (TiO_2_), have been reported in the literature due to their higher affinity and electrocatalytic activity, which enhances the detection limit of OP compounds. The nanostructures of metal oxides and hydroxides can exhibit better redox activity at the electrode surface in the detection of OP pesticides due to their intrinsic large surface areas and higher affinity by providing higher selectivity and sensitivity with lower detection limits (LOD). Liu and Lin have reported a novel disposable screen-printed portable electrochemical sensor developed by a ZrO_2_ nanoparticle-deposited Au electrodes for the detection of nitroaromatic OPs [44]. Gong et al. developed a highly sensitive sensor using a glassy carbon electrode modified with zirconia nanoparticles decorated with graphene nanosheets [73]. The strong selective affinity of zirconia for the phosphoric group and high surface area cause it to quantify the electroactive OP compounds in a facile manner. Further, the basic and acidic surface-active sites and abundant surface oxygen crystal defects of zirconia provide higher electron transfer processes and ion exchange capacity [74]. Very recently, CuO-nanorod electrodes synthesized by the anodization of Cu foil and annealing was able to elevate the sensitivity of four different types of OPs, as these one-dimensional nanostructures can provide a desirable electrocatalytic performance and strong affinity towards OPs [75]. Mixed metal oxide nanostructures such as CuO-TiO_2_ nanocomposites [76] or metal oxide nanocomposites such as CuO NP/three-dimensional (3D) graphene nanocomposite [77] have become more popular than metal oxide nanostructure-based sensors in the detection of OPs. The synergic effect of dual or many materials in the nanocomposite can improve the performance of the electrochemical sensor. Figure 3a represents the working principle of the fabricated sensor; the morphology of the CuO NPs that was fabricated on 3D graphene to detect malathion; and the performances of the electrochemical sensor towards malathion with the presence of common inorganic ions, glucose, and other pesticides such as carbendazim (Car), lindane (Lin), or trichlorphon (Tri). This study demonstrates the capability of this electrode to be used for the detection of real samples, with excellent anti-interference performances.

Apart from metallic nanomaterials, non-metallic nanostructures play a major role in fabricating electrochemical sensors. Graphene is one of the more interesting carbonaceous nanomaterials and is a popular electrode material with unique and fascinating physicochemical properties. The exceptional electrical conductivity, good mechanical stability, high surface area, and absorption capacity of graphene provide the improved performances of electrochemical sensing [69,73,78]. The different derivatives of graphene (i.e., graphene oxide, reduced graphene oxide(rGO), graphene nanoribbons, graphene, quantum dots) have different electronic characteristics and properties. Therefore, more recently, graphene nanoribbons and nanoplatelets have been reported for sensing applications with better performances compared to pristine graphene nanosheets due to their rich dispersion and surface area to provide higher electrical conductivity [79,80]. Another carbon-based non-metallic nanomaterial is one-dimensional CNTs, which have extremely promising properties, including excellent mechanical strength, electrical conductivity, chemical stability, and high surface area. Both graphene and CNT interact via π-π interactions and hydrophobic interactions with OPs, providing excellent binding affinity to detect OPs. Several examples of nanocomposites are made up of these carbon-based nanomaterials as they offer excellent properties. Both single-walled carbon nanotubes (SWCNTs) and multi-walled carbon nanotubes (MWCNTs) [81] can enhance the electrochemical sensing performance, having high electrical conductivity due to their one-dimensional electronic pathways to accumulate effective charge transportation to electrodes and large surface area [81,82]. Other materials, such as bismuth films, polymers, and silicon carbide (SiC) have also been seen to contribute to the electrochemical performances of OPs [83,84,85]. The abundance of these nanomaterials and their cost in terms of their properties controls the selection of these materials in electrochemical sensors. Zhao et al. have reported a new nanocomposite made up of halloysite (Hal) nanotubes/MWCNTs to detect methyl parathion (MP) pesticide [86]. They have explained the contribution of the higher number of active sites and nanopores on the surface for effective charge transport with assistance from a one-dimensional hallow halloysite aluminosilicate clay mineral that has both silicate and aluminum hydroxide groups (Si-O-Si and Al-OH). Figure 3b represents the Hal/MWCNTs nanocomposite fabrication process and the electrochemical detection process of methyl parathion on the prepared electrode.

**Table 2 sensors-23-01226-t002:** Summary of enzyme-free nanomaterial-based electrochemical sensors utilized for organophosphorus compounds detection.

Nanomaterials/ Electrode	OP Compound	Detection Technique	Concentration Range	LOD	Sensitivity	Published Year
**ZrO_2_ NPs**	Paraoxon, fenitrothion, Methyl parathion	Square wave voltammetry	Methyl parathion: 5–100 ng/mL	3 ng/mL	-	2005 [44]
**ZrO_2_/Au nanocomposite**	Parathion	Square wave voltammetry	10–160 ng/ml	3 ng/ml	-	2008 [70]
**MWCNTs**	Methyl parathion	Square wave voltammetry	0.05–2.0 µg/mL	0.005 µg/mL	-	2008 [87]
**Bismuth-film/GCE**	Methyl parathion	Square wave voltammetry	3.0–100 ng/mL	1.2 ng/ml	0.0253 µA.mL.ng^−1^	2008 [83]
**ZrO_2_ NPs**	Methyl parathion	Square-wave voltammetry	0.003–2.0 µg/mL	0.001 µg/mL	-	2008 [88]
**Ionic liquid/** **SWCNT/GCE**	Methyl parathion	Linear sweep voltammetry	2.0 × 10^−9^–4.0 × 10^−6^ M	1.0 × 10^−9^ M	-	2008 [81]
**Electro deposited Au NPs on MWCNTs**	Parathion	Linear scan voltammetry	6.0 × 10^−5^–5.0 × 10^−7^ M	1.0 × 10^−7^ M	-	2009 [89]
**ZrO_2_ NPs modified carbon paste electrode**	Methyl parathion	Square wave voltammetry	5.0–3000.0 ng/mL	2.0 ng/mL	-	2010 [90]
**Pd NPs/MWCNTs nanocomposite**	Methyl parathion	Differential pulse voltammetry	0.10 μg/mL– 14 μg/mL	0.05 μg/mL	-	2010 [71]
**Au NPs/Nafion/GCE**	Methyl parathion	Square wave voltammetry	5.0 × 10^−7^ to 1.2 × 10^−4^ M	1.0 × 10^−7^ M	-	2010 [91]
**ZrO_2_/carbon paste electrode**	Methyl parathion	Square wave voltammetry	1.0 × 10^−8^– 1.0 × 10^−5^ mol/L	4.6 × 10^−9^ mol/L	-	2010 [92]
**Au NPs decorated** **graphene hybrid nanosheets**	Methyl parathion	Square wave voltammetry	0.001–0.1 & 0.2–1.0 µg/mL	0.6 ng/mL	-	2011 [69]
**Graphene-ZrO_2_ nanocomposite**	Methyl parathion	Square wave voltammetry	0.5 ng/mL–100 ng/mL	0.1 ng/mL	-	2011 [93]
**Au NP-MWCNTs composite**	Methyl parathion	Differential pulse voltammetry	0.5–16.0 mg/mL	50 µg/mL	1.91 μAμg^−1^mL	2011 [94]
**ZrO_2_ NPs-graphene nanosheet nanocomposite**	Methyl parathion	Square wave voltammetry	0.002–0.9 µg/mL	0.6 ng/mL	-	2012 [73]
**Au–ZrO_2_–SiO_2_** **nanocomposite**	Paraoxon-ethyl	Square wave voltammetry	1.0–500 ng/mL	0.5 ng/mL	-	2012 [95]
**MWCNTs/poly(acrylamide) nanocomposite/GCE**	Methyl parathion	Differential pulse voltammetry	5.0 × 10^−9^– 1.0 × 10^−5^ mol/L	2.0 × 10^−9^ mol/L	0.882 μA.μM^−1^	2012 [96]
**Graphene-Nafion/GCE**	Methyl parathion	Square wave voltammetry	0.02–20 µg/mL	1.6 ng/mL	-	2013 [97]
**MWCNTs-CeO_2_-Au nanocomposite**	Methyl parathion	Electrochemical stripping voltammetry	10^−10^ × 10^−7^ M	3.02 × 10^−11^ M	-	2013 [98]
**CuO NWs-SWCNTs nanocomposite**	Malathion	Differential pulse voltammetry	0-0.4 ppb	0.1 ppb	628.71 μA.cm^−2^.ppb^−1^	2014 [37]
**ZrO_2_ NPs**	Omethoate	Square wave voltammetry	98.5 pmol/L–985 nmol/L	52.5 pmol/L	-	2015 [99]
**mono-6-thio-b-cyclodextrin/** **Au NPs/SWCNT/** **GCE**	Methyl parathion	Square wave voltammetry	2.0–80.0 nM	0.1 nM	0.035 μA/nM	2015 [100]
**Reduced graphene oxide/AuNPs nanocomposite**	Fenitrothion	Differential pulse voltammetry	0.1–6.25 ng/mL	0.036 ng/mL	-	2016 [101]
**Ag/graphene nanoribbons nanocomposite**	Methyl parathion	Amperometry	0.005–2780 µM	0.5 nM	0.5940 μA.μM^−1^ cm^−2^	2017 [72]
**MoS_2_/graphene nanocomposite**	Methyl parathion	Amperometry	0.01–1905 µM	3.23 µM	0.457 (±0.008) μA.μM^−1^.cm^−2^	2017 [102]
**Graphene oxide/PEDOT: PSS & polypyrrole/AuNPs**	Malathion, cadusafos	Impedance spectroscopy	N/A	0.1 nmol/L	-	2017 [84]
**3 D graphene/AuNPs nanocomposite**	Diethylcyanophosphone	Differential pulse voltammetry	1 × 10^−11^–7 × 10^−8^ M	3.45 × 10^−12^ M	-	2017 [103]
**MWCNTs/TiO_2_ NP/** **GCE**	Diazinon	Square wave voltammetry	11–8360 nM	3 nM	-	2017 [82]
**NiO nanoplatelets**	Parathion	Differential pulse voltammetry	0.1–30 µM	0.024 µM	-	2018 [25]
**CuO NPs/** **3 D graphene** **nanocomposite**	Malathion	Differential pulse voltammetry	0.03–1.5 nM	0.01 nM	-	2018 [77]
**CuO/TiO_2_ nanocomposite**	Methyl parathion	Differential pulse voltammetry	0–2000 ppb	1.21 ppb	-	2018 [76]
**Pralidoxime chloride** **immobilized CuO** **nanostructure**	Chlorpyrifos, fenthion, methyl parathion	Differential pulse voltammetry	0.01–0.16 µM	Chlorpyrifos-1.6 × 10^−9^ M Fenthion-2.5 × 10^−9^ M Methyl parathion-6.7 × 10^−9^ M	-	2018 [104]
**NbC/Mo NPs nanocomposite**	Fenitrothion	Differential pulse voltammetry	0.01–1889 µM	0.15 nM	0.355 μA.μM^−1^ cm^−2^	2018 [105]
**Peptide nanotube/** **pencil graphite electrode**	Fenitrothion	Square wave voltammetry	0.114 μM–1.712 μM	0.0196 μM	-	2018 [106]
**Ionic liquid/chitosan/AuNPs**	Malathion	Square wave voltammetry	0.89–5.94 nM & 5.94–44.6 nM	0.68 nM	-	2018 [38]
**AuNPs/neutral red-protein functionalized graphene/GCE**	Methyl Parathion	Differential pulse voltammetry	0.02–0.153 μM & 0.153–1.36 μM	6 nM	-	2019 [107]
**SiC NPs/MWCNTs/t chitosan**	Parathion	Differential pulse voltammetry	0–10,000 ng/mL	20 ng/mL	-	2019 [85]
**CNT/carbon paste electrode**	Diazinon	Differential pulse voltammetry	1 × 10^−10^–6 × 10^−8^ M	4.5 × 10^−10^ M	-	2019 [108]
**MWCNTs/ZrO_2_ nanocomposite**	Methyl parathion	Differential pulse voltammetry	19.9–176.8 × 10^−6^ mol/L	9.0 × 10^−9^ mol/L	-	2020 [74]
**MWCNTs**	Methyl parathion	Differential pulse voltammetry	1.0 × 10^−7^−3.4 × 10^−5^ M	3.52 × 10^−8^ M-acid treatment, 3.33 × 10^−8^ M-base treatment	0.55 μA/μM-acid 0.52 μA/μM-base	2020 [109]
**Pt/Zr-based metal organic framework/carbon paste microelectrode**	Phosalone	Square wave voltammetry	0.50 nM–20 μM	0.078 nM	-	2020 [110]
**Carbon nanoballs/Glove-Right finger pretreated**	Fenitrothion	Square wave voltammetry	1.0 × 10^−6^– 10 × 10^−6^ mol/L	6.4 × 10^−7^ mol/L	-	2021 [111]
**Graphene nanoplatelet/** **ZrO_2_/** **Ag SPE**	Methyl parathion	Square wave voltammetry	1–20 µM	1 µM		2021 [112]
**Zinc(II) phthalocyanine/** **SWCNT/GCE**	Methyl parathion	Differential pulse voltammetry	2.45 nM–4.0 × 10^−8^ M	1.49 nM	0.1847 μA.nM^−1^	2021 [113]
**CuO Nanorods electrode**	chlorpyrifos, parathion, paraoxon, pirimiphos	Cyclic Voltammetry	0.29–0.61 μM	10^−7^ M	1.269 μA/ngmL^−1^ -chlorpyrifos, 1.425 μA/ngmL^−1^ -parathion, 1.657 μA/ngmL^−1^ -paraoxon, 2.833 μA/ngmL^−1^ -pirimiphos	2021 [75]
**Halloysite nanotubes/MWCNTs**	Methyl parathion	Differential pulse voltammetry	0.5–11 μM	0.034 μM	-	2021 [86]
**Ag NPs/PGE functionalized CNT/g** **raphite electrode**	Diazinon, malathion, chlorpyrifos	Multiple pulse amperometry	Diazinon: 0.1–20 μM, malathion: 1–30 μM, chlorpyrifos: 0.25–50 μM	Diazinon-0.35 μmol/L, malathion-0.89 μmol/L, chlorpyrifos-0.53 μmol/L	Diazinon- 0.068 mALμmol^−1^, Malathion-0.030 mALμmol^−1^ Chlorpyrifos-0.043 mALμmol^−1^	2022 [114]

**Figure 3 sensors-23-01226-f003:**
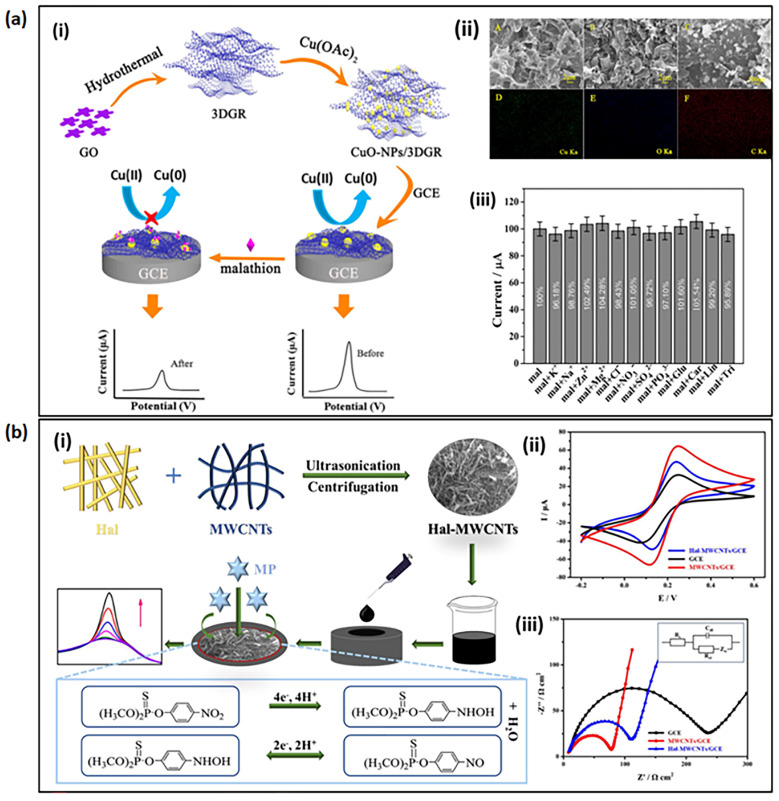
(**a**) (i) Schematics of the electrochemical detection of malathion using the fabricated CuO-NPs/3D graphene/glassy carbon electrode (GCE). (ii) Scanning electron microscopy (SEM) images of graphene, CuO-NPs/3DGR nanocomposite and CuO-NPs/3D graphene nanocomposite with their elemental mappings (iii) The bar graph that shows the selectivity of CuO-NPs/3D graphene/GCE without and with different interferences at a fixed concentration of malathion (1 nM). These images were adapted from Xie et al. [77], with permission from Elsevier. (**b**) (i) Schematics of the fabrication process of the Hal-MWCNTs/GCE electrochemical sensor and its mechanism to detect methyl parathion. (ii) Cyclic Voltammetry (CV) curves. (iii) Electrochemical Impedance Spectroscopy (EIS) plot of Hal-MWCNTs/GCE electrode compared to bare GCE and MWCNTs/GC electrodes. These images were adapted from Zhao et al. [86] with permission from Elsevier.

Molecular imprinted polymers (MIPs) are another promising candidate for the electrochemical sensing of OPs due to their simplicity, cost effectiveness, thermal resistance, and chemical stability, and a few recent studies have reported the detection of fenamiphos [115], parathion [116,117], and methyl-parathion [118,119,120] pesticides. By choosing the most suitable monomers, MIPs can create their specific cavities designed for the template molecule through the polymerization of monomers with the aid of a cross-linker and an initiator. The selectivity is targeted using the template regarding size, shape, and position of the functional group of binding OPs [121]. However, the selection of a proper monomer and a suitable synthesis protocol to tailor the specificity and selectivity of the analytes are major challenges for their use for real-time sensors, and those parameters need to be optimized to achieve the best performances. 

The introduction of novel functional nanomaterials or composites is essential to overcome the challenges of electrochemical sensors, such as to obtain low LOD, sensitivity, selectivity, reproducibility, and stability in real samples analysis. Nanomaterial-based sensors have achieved ultra-sensitivity with selectivity due to the enhanced electro catalytic activity [122,123]. Every nanomaterial has its own intrinsic properties, associated with their structure, size, functionality, and morphology. The common merits of nanomaterials are high conductivity, accelerated electron transfer and larger surface area, and a high selective affinity towards OPs [124,125]. The design of a sensor to achieve better features of the electrochemical sensor is challenging, and it requires a good nanomaterial selection that is compatible with the electrode substrate [126]. The knowledge of the guiding principles of controlled fabrication of nanomaterials to control these properties over different synthesis strategies is lacking [127]. However, it is still not common to find attempts that have addressed the essential features of a sensor that is suitable for environmental samples.

### 2.3. Molecular Electrochemical Sensors 

To date, few small molecules have been used to fabricate electrochemical sensors other than different nanomaterials, which are stated in Table 3. Among molecules, one of the interesting functional materials is metallophthalocyanines, which provide highly selective and highly chemically and thermally stable sensors. This class of materials contain redox active metal centers and electrochemically active substituents. Various studies have reported metallophthalocyanines as OP pesticides and other pesticide sensors by using different metal cations to tailor the cavity size and porosity to influence the sensing capabilities [118,119,120]. These molecule-based sensors also have good limits of detection range with good selectivity due to their highly conjugated structure. As they have strong absorbance in the UV–vis region, they can be used as photocatalysts and photoconductors [128]. The coatings of metallophthalocyanine films can be fabricated by different thin film deposition techniques, including physical vapor deposition (PVD), spin coating, Langmuir–Blodgett technique, electropolymerization, electrodeposition, or covalent anchoring. 

Atıf Koca et al. developed an electrochemical pesticide sensor using a redox richness cobalt phthalocyanine–anthraquinone (CoPc-AQ) hybrid complex for the detection of carbofuran and eserine pesticides by increasing the selectivity of the sensor [129]. Immobilization of nano-platinum on the ITO/CoPc-AQ films achieved higher sensitivity with a lower detection limit of 2.30 × 10^−9^ M. Another study fabricated three new electrodes based on cobalt(II), titanium(IV), and manganese(III) phthalocyanines to alter the redox activity of the electropolymerized films to investigate their selectivity and sensitivity, as shown in Figure 4 [130]. Those prepared electrodes were tested for chlorpyrifos, fenitrothion, and methomyl pesticides, and this study has demonstrated that the change of the substituents can alter the sensing and selectivity of OP compounds significantly.

**Table 3 sensors-23-01226-t003:** Summary of enzyme-free molecule-based electrochemical sensors utilized for organophosphorus compounds detection.

Molecule/ Electrode	OP Compound	Detection Technique	Concentration Range	LOD	Sensitivity	Published Year
**Cobalt metallophthalocyanines/GCE**	Fenitrothion	Square wave voltammetry	1.20–42.0 μmoldm^−3^	0.460 μmoldm^−3^	0.26 Acm^−2^M^−1^	2017 [130]
**CoPc(MOR-NAF)/GCE**	Diazinon, Parathion	Square wave voltammetry	Diazinon-0.38–5.07 μmoldm^−3^, Parathion-0.07–5.75 μmoldm^−3^	Diazinon-0.120 Μmoldm^−3^ Parathion-0.020 μmoldm^−3^	3.46 Acm^−2^M^−1^	2017 [131]
**Manganese** **Phthalocyanine-4-azido polyaniline hybrid/ITO**	Fenitrothion, eserine, diazinon	Square wave voltammetry	Fenitrothion-0.12–15.00 μmoldm^−3^ Eserine-0.10–5.00 μmoldm^−3^ Diazinon-0.20–7.50 μmoldm^−3^	Fenitrothion-0.049 μmoldm^−3^ Eserine-0.088 μmol dm^−3^ Diazinon-0.062 μmol dm^−3^	4.67 Acm^−2^M^−1^	2019 [132]

## 3. Flexible Electronic Sensors

Developing economically viable methods to integrate sensors into flexible and portable devices is an effective and rapid way of pesticides detection in real samples. A few popular approaches are shown in Figure 5. Lithographic approaches are one of the popular techniques of the scalable manufacturing of sensors. High spatial resolution can be achieved through this method, which enables high density and miniaturization [133]. A few different types of lithography techniques are electron lithography, photolithography, hard lithography, soft lithography, and nanoimprint lithography [134,135,136,137]. However, it is necessary to balance the economic cost of the sensor with the performances, such as resolution and throughput. Our group has developed a photolithographic gold interdigitated sensor for the detection of OP compounds, as shown in Figure 5a [46]. In terms of disposable electrochemical sensors, the choice of the substrate and the methodologies of the electrode construction are the two main concerns and are employed in various scales. An underlying soft substrate and conductive electrode materials are needed to obtain a flexible sensor. This selection is the key factor in achieving the device’s performance and flexibility [138,139]. The adhesion between the substrate and the electrode material depends on their nature and directly affects the sensor’s overall stability. The complementary energy surface of these two components drives the homogeneity in the sensor [138].

The electrode modification process can also be performed using a screen-printing technique (SPT), which is a low-cost method for mass production (Figure 5b) [140,141,142,143,144]. Such electrode modification using different nanomaterials [25,145,146,147] has advanced the development of flexible electronics. Furthermore, nanometer thickness coatings provide more catalytic activity due to their high surface-to-volume ratio. The ink’s preparation and penetration to the substrate are crucial to achieve the homogeneity of the sensor. The choice of solvent with a low vapor pressure and boiling point to dissolve the traces of conductive ink is important to have better flowability and adsorb the substrate [138,148]. Another SPT method challenge is using ink binders. The mixture of active electrode material, binders, stabilizers, or additives can cause the sensor’s poor electron transportation if they are incompatible [149]. Although it is a challenge to eliminate the binder and additives completely, the amount can be potentially reduced to improve the adhesion and wettability of the ink. A few examples of nanocomposite-modified SPT-based electrodes have been reported for the detection of OP residues in vegetables and fruit samples [72,105]. Thota and Ganesh have reported a novel disposable, flexible chemically modified overhead projector strip-based electrode for the detection of MP. It is a good extension to attempt the selective and sensitive flexible integration of electrodes by using low cost and nonconductive sheets [51].

Stamp transfer is an alternative fabrication method to SPT, comprising both conductive and insulating inks on a flexible substrate using a pattern-transfer technique [150]. Stamp transfer offers various advantages for making wearable devices, as this technique is highly compatible with irregular substrates such as skin [150,151]. Three-dimensional printing and inkjet printing are other additive manufacturing techniques for making flexible sensors through a digitally controlled layer-by-layer deposition process [152]. They have been classified as non-template methods as the ink can be distributed freely on the surface of the substrate [151]. These methods are more compatible with various substrates than other techniques. They offer many advantages, such as freedom to work with different geometries and substrates, high accuracy and durability, repeatability, and low cost [152,153,154]. Preliminary studies of 3D printed electrodes consisting of nanocarbon and polylactic acid for the detection of four OPs (fenitrothion, methyl parathion, parathion, and paraoxon) have been reported elsewhere [155]. However, these methods are still in their preliminary stages in in-situ monitoring techniques, and only a few attempts at portable electrochemical sensors have been reported in the literature. 

Roll-to-roll processing is another recent fabrication method of sensors that builds different functional structures on a substrate material at an industrial scale at a low cost. Although there are a few reports on roll-to-roll manufacturing for the detection of pesticides and other materials [156,157,158], only one study has investigated the sensing of OPs (Figure 5c) [112]. Stanciu et al. manufactured a graphene nanoplatelets/ZrO_2_ NPs-based roll-to-roll enzyme-less electrochemical sensor to detect nitroaromatic OP pesticides [112]. The advantage of the synergic effect of zirconia’s electrocatalytic and adsorption efficacy towards phosphates and graphene’s higher conductivity could achieve a high sensitivity for methyl parathion and fenitrothion pesticides. This type of high-volume fabrication of flexible portable electronic devices enables an effective and efficient way of on-site monitoring of OPs in the environment. Compared to traditional rigid sensors, flexible electrochemical sensors are mostly made up of hybrid nanocomposites composed of both inorganic and organic materials to address both the flexibility and sensitivity of the sensor. The nanomaterial composition determines the sensor’s quality, and it is a great challenge to overcome the limitations of material properties and fabrication methods [159]. Future work is needed to fabricate enzyme-free electrochemical sensors as wearable point-of-use screening tools towards a wide range of applications, including health, defense, and food security.

**Figure 5 sensors-23-01226-f005:**
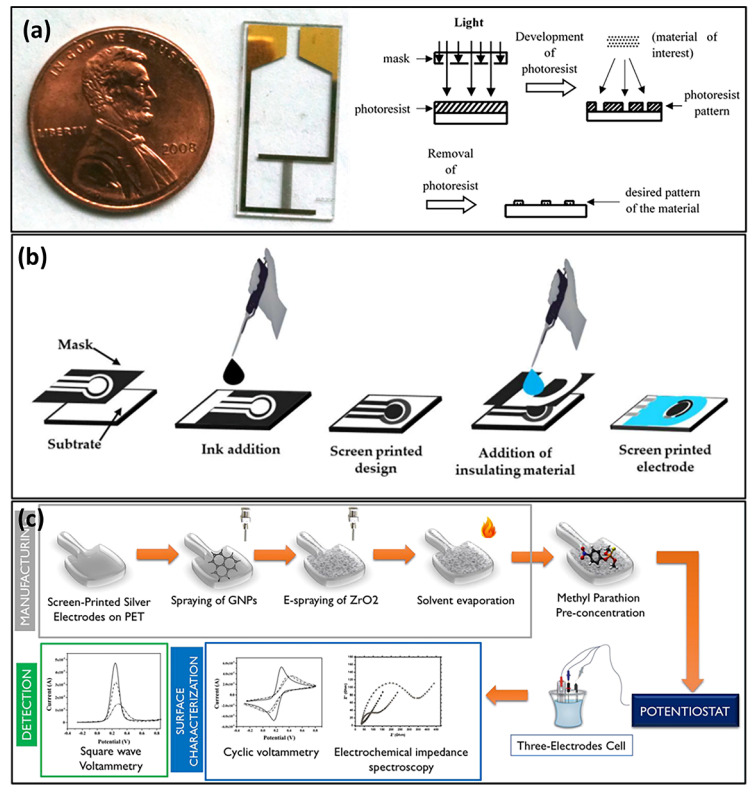
(**a**) Photolithographically-fabricated electrochemical sensor and schematic of the fabrication process. These images were adapted from Narakathu et al. [46], with permission from Elsevier and Park and Shuler [160] and with permission from John Wiley and Sons, respectively. (**b**) Manufacturing stages of SPEs [142], reproduced with permission from Pérez-Fernández et al., published by MDPI, 2020. (**c**) A fully roll-to-roll manufactured electrochemical sensor based on graphene nanoplatelet and ZrO_2_, reproduced with permission from reference [112], Copyright {2021} American Chemical Society.

## 4. Summary, Challenges, and Future Perspectives

During the last century, organophosphorus compounds have been used worldwide to increase crop production. However, their residues in the environment are very toxic to non-target species, including humans and mammals. Therefore, effective technologies are reported in the literature to accurately determine these OP pesticides. Electrochemical methods are the most popular technologies to detect electroactive compounds such as OP pesticides using different techniques such as voltammetry, amperometry, and potentiometry. Recently, developing enzyme-free sensors has gained more interest, particularly by using different nanomaterials and molecules to fabricate portable sensors with higher sensitivity, selectivity, and durability by overcoming the limitations of enzyme-based sensors.

Commercialization and large-scale manufacturing strategies for the development of electrochemical sensors are the fastest developing areas in contemporary environmental technology due to the increased awareness of global environmental and health impacts. However, the technology of the real-time sensing of OPs is still in its infancy. During the past few decades, different approaches for the fabrication of electrochemical sensors have been proposed using various novel nanomaterials and molecules without the support of enzymes. The excellent physical, chemical, and electronic properties of different materials were utilized to design these sensors, incorporating enhanced stability and sensitivity. The most reported electrochemical sensors are made up of two or multiple hybrid materials to promote different properties of the sensor. Moreover, multi-detection abilities also can be addressed by having hybrid materials for real environmental applications. By incorporating both electrochemical and optical sensors to build a dual sensor it is possible to further enhance the applicability of the sensor by minimizing the false positive signals. There are only a few preliminary studies aimed at addressing the portable disposable sensors for rapid on-site detection of OPs in agriculture, food, health, and military fields. Such platforms are necessary to facilitate next generation human health problems for their better life. 

Although previous works have already laid out the background to initiate the development of rapid, highly sensitive, reliable, and facile electrochemical sensors for OP detection, certain areas still need to be addressed to achieve rapid monitoring systems that can be integrated into flexible electronics. One of the main challenges of using nanostructured materials for the electrochemical sensors is surface-fouling at the surface of the electrodes due to the unnecessary binding affinity with interferents in wastewater, which causes the limited lifetime of electrochemical sensors. This issue should be addressed by designing antifouling strategies, with a modified electrode or material that has antifouling properties (i.e., metallic nanostructures). Furthermore, this issue can be overcome by developing disposable strips for sensors. 

Successful determination of OP in the polluted wastewater sample is complicated due to the large interferences of small ions and molecules, and the selective detection of OPs is still a great challenge. Although there are few studies on the binding mechanism of target OPs, more scientific investigations are greatly needed. The insight of the interaction between the electrode surface and OPs at an atomic level is still lacking and needs to be addressed. Toward the integrating technologies of portable sensors, large manufacturing strategies have challenges associated with stability and repeatability. In the future, the feasibility study on the industrialization of both electrochemical sensors should be strengthened. Developing novel electrochemical sensors is important to overcome the challenges of detection limits with higher selectivity of different OP compounds. These methods are limited to a few nanomaterial types, and very few molecules have been introduced to fabricate OP sensors. Array technology would be a key remedy to avoid signal alteration due to the multiple pesticides in the sample mixture. However, it is still in an embryonic stage. Therefore, more research work is needed in order to develop enzyme-free sensing devices to determine OP pesticides due to their adverse effects on human health.

Portability, flexibility, and wearability are the key features of emerging electrochemical sensors for the detection of OP pesticides. Nanomaterials have made an effective contribution to the integration of flexible electrochemical sensors for real-time monitoring devices. Improved sensitivity, selectivity, and sensitivity can be achieved by introducing novel hybrid materials. The future development of flexible sensors needs to carefully select these nanomaterials in terms of their abundance, lower toxicity, or biodegradability and cost effectiveness. This can minimize the risk to individuals in various fields such as health, food, military, forensics, and ecology. Although these sensors still only focus on health applications, they need to expand to all fields. However, the tremendous growth of different types of disposable sensors, including electrochemical sensors, has now raised awareness of environmental pollution. Electronic waste (E-waste) is a current critical issue that needs urgent attention to preserve the environment. Disposable sensors mostly consist of standard materials for micro- and nanoelectromechanical systems, synthetic polymers, cellulose-based materials, or hybrid materials. If these fabrication materials have properties such as recyclability, biodegradable, or even composability, this can address the issue of environmental sustainability. However, it does not mean that all these materials need to be biodegradable. The introduction and use of novel greener, ecofriendly materials and low waste fabrication strategies would strengthen the mitigation strategies to reduce the adverse environmental impacts of disposable electrochemical sensors. 

## Figures and Tables

**Figure 1 sensors-23-01226-f001:**
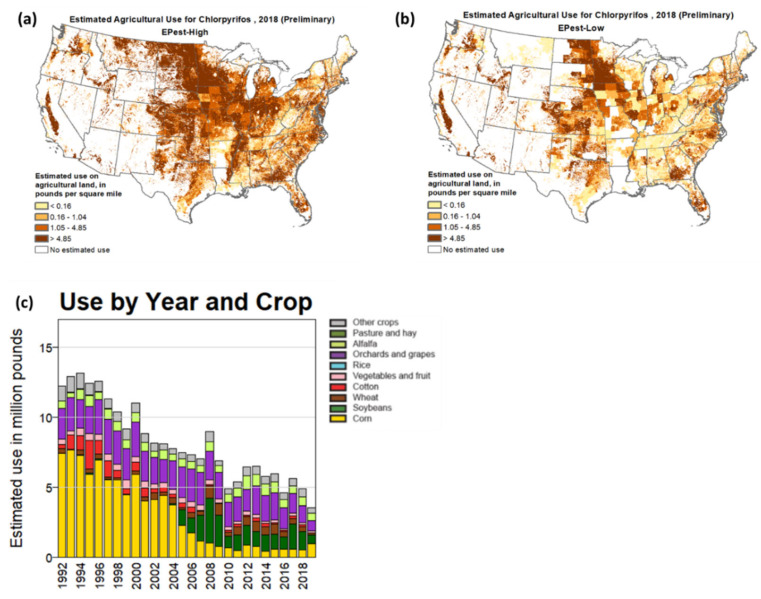
The map of estimated agricultural usage of chlorpyrifos in 2018: (**a**) EPest-high; (**b**) EPest-low; and (**c**) estimated usage for different crops from 1992 to 2018 [6].

**Figure 2 sensors-23-01226-f002:**
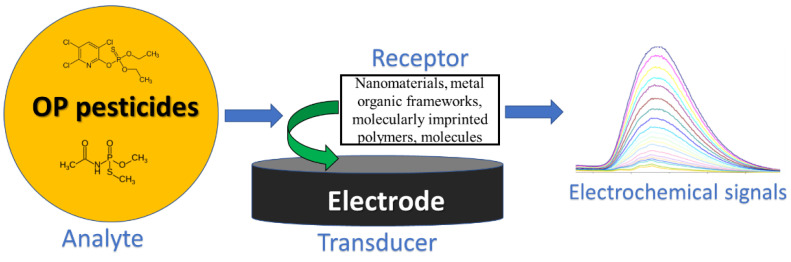
Schematic diagram of the non-enzyme-based electrochemical sensor.

**Figure 4 sensors-23-01226-f004:**
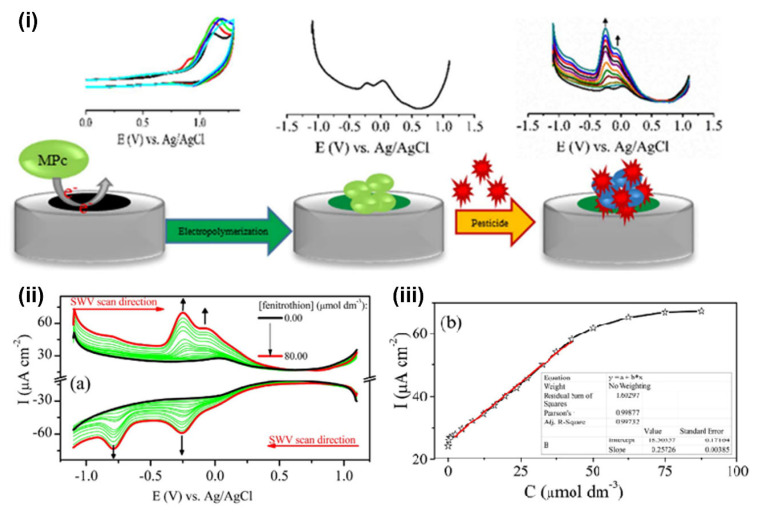
(**i**) Schematics of the electrochemical sensor fabricated by electropolymerization of metallophthalocyanines. (**ii**) SWV responses of GCE/CoPc(ma) during the titration with fenitrothion pesticide and (**iii**) its calibration line. These images were adapted from Akyüz et al. [130] with permission from Elsevier.

**Table 1 sensors-23-01226-t001:** Chemical structures of recently used common OP insecticides in the US (source: U.S. EPA data, 2008–2012) [3].

OP Compound	Chemical Structure	Usage Range in 2012 (Millions of Pounds)
chlorpyrifos	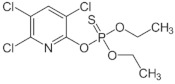	5–8
acephate	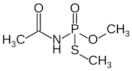	5–8
malathion	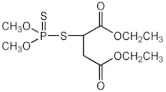	1–4
naled	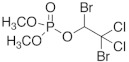	1–2
phorate	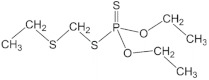	1–2
dicrotophos	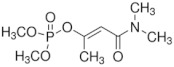	1–2
dimethoate	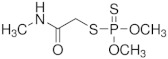	<1
phosmet	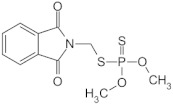	<1
ethoprophos	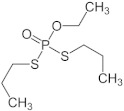	<1

## Data Availability

Not applicable.

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
