# Peer review of "Recent Advances of Enzyme-Free Electrochemical Sensors for Flexible Electronics in the Detection of Organophosphorus Compounds: A Review"

_sensors, 2023, doi:10.3390/s23031226_

Round 1

Reviewer 1 Report

This manuscript provides a comprehensive overview of non-enzyme based electrochemical and optical sensors. The review discusses the remarkable advancements in the fabrication of various novel materials without the support of enzymes for electrochemical, optical, and/or dual sensors. The sensing mechanisms, synthesis methods, current state of the art flexible electronic sensors developed using nanomaterials for organophosphorus pesticides detection are described in detail. This review paper is well written and I recommend its publication in the Sensors journal.

1.         The authors should replace Figure 3 and 4 with high resolution images.

2.         The full forms of Ag, Au, Pd, NiO, CuO and TiO2, ZrO2 were not provided when they were mentioned for the first time (Line 195, 196, 200, 206).

3.         Please improve the references section by citing the following publications.

·         Saeed et al. (2019). Synthesis of a novel hexaazatriphenylene derivative for the selective detection of copper ions in aqueous solution. RSC advances, 9(68), 39824-39833. DOI: 10.1039/C9RA08825C

·         Thota, R., & Ganesh, V. (2016). Selective and sensitive electrochemical detection of methyl parathion using chemically modified overhead projector sheets as flexible electrodes. Sensors and Actuators B: Chemical227, 169-177.

·         Narakathu, et al. (2011). Detection of picomolar levels of toxic organophosphorus compounds by electrochemical and fluorescence spectroscopy. Sensor Letters, 9(2), 907-909. DOI: https://doi.org/10.1166/sl.2011.1641

·         Avuthu et al., (2019). A Screen Printed Phenanthroline-Based Flexible Electrochemical Sensor for Selective Detection of Toxic Heavy Metal Ions. IEEE Sensors Journal, vol. 16, no. 24, pp. 8678-8684, 15 Dec.15, 2016, doi: 10.1109/JSEN.2016.2572184.

Reviewer 2 Report

This manuscript reviews the flexible electrochemical sensors for the detection organophosphorus compounds. However, the review for the electrochemical sensors including flexible electrodes for organophosphorus compunds detection has been reported (TrAC-Trend. Anal. Chem., 2017, 92, 62-85; TrAC-Trend. Anal. Chem., 2020, 132, 116041; J. Adv. Res., 2022, 37, 61-74). The introduction with limited length for the flexible electrochemical sensors is not well presented. For example, the advantage and significance of flexible electrodes are not well discussed. And the key role of nanomaterials in construction of flexible sensors especially the sensing property still needs to be discussed in detail. Moreover, the other important factor in the construction of flexible detection electrodes is missed in the manuscript. Hence, I cannot recommend the publication of this manuscript in the journal of Sensors. Before submitting it to the other journal, the authors need to consider the points below:

1. The advantage of nanomaterials especially in the construction of flexible electrodes needs to be well presented. And the introduction of electrochemical sensor can be carried out based on the kind of materials such as metallic nanomaterials and non-metallic nanomaterials. And the advantage and shortage of applied nanomaterials can be provided in this part.

2. The key role of nanomaterials in the electrochemical detection towards sensitivity, selectivity and stability needs to be well discussed.

3. The methods and strategies for the fabrication of flexible electrochemical sensors needs to be emphatically introduced in the manuscript. Other techniques such as stamp transferring and 3D printing are suggested to be added.

4. The introductions of significant factor such as substrates, binder agent and solvents for the construction of flexible electrochemical sensors needs to be added in the manuscript.

5. The significance of enzyme-free electrochemical sensor needs to be presented in Title.

6. The requirement or advantage of flexible sensing electrodes is suggested to be well discussed in Introduction.

7. The wearable glove and wireless sensors for detection of organophosphorus compounds is suggested to be added in the manuscript.

8. The application of flexible electrochemical sensor for organophosphorus pesticide residues in real environmental and food samples can be further introduced in the manuscript.

9. The perspectives for the improvement of nanomaterials in flexible electrochemical sensors to enhance its detection property is suggested to be added in the part of “6. Summary, challenges, and future perspectives”.

10. The format of references needs to be revised and further checked.

Reviewer 3 Report

The manuscript needs minor revisions.

Reviewer 4 Report

This review article focuses on the recent advances, mainly in materials perspectives, of enzyme free electrochemical sensors. The topic is interesting for the common readers as well those working in the field of sensors. Overall, the manuscript is well-written and can be considered for publication after minor revision. Some of the points for revision is given as under:

1. The flexible electronic sensor part is very short and not comprehensive. It should be reviewed more in detail and devices if developed should be presented.

2. Optical sensors are coupled with Flexible sensors. This should be separately discussed more in detail and the results should be presented in a figure.

3. Environmental concerns associated with disposing off the sensors should be addressed and mitigation strategies may be presented.

Round 2

Reviewer 2 Report

The manuscript has been revised based on the comments from reviewers. The discussion of methods as Stamp transfer and 3D printing for preparation of flexible electrochemical sensor as well as the factors as substrates, binder agent and solvents for fabrication of flexible electrode have been added in the manuscript. Besides, the construction and application of novel hybrid nanomaterials with environmental sustainability has been proposed as a strategy for improving the property of flexible electrochemical sensors in part of “4. Summary, challenges, and future perspectives”. Furthermore, the introduction and title have been revised to clearly illustrate the advantages of flexible enzyme-free electrochemical sensor based on nanomaterials. In addition, the figures, tables and references have also been improved. Hence, this paper can be accepted for publication in “Sensors.